# Molnupiravir and Its Active Form, EIDD-1931, Show Potent Antiviral Activity against Enterovirus Infections In Vitro and In Vivo

**DOI:** 10.3390/v14061142

**Published:** 2022-05-25

**Authors:** Yuexiang Li, Miaomiao Liu, Yunzheng Yan, Zhuang Wang, Qingsong Dai, Xiaotong Yang, Xiaojia Guo, Wei Li, Xingjuan Chen, Ruiyuan Cao, Wu Zhong

**Affiliations:** 1National Engineering Research Center for the Emergency Drug, Beijing Institute of Pharmacology and Toxicology, Beijing 100850, China; lyx1986528@126.com (Y.L.); lmm200821@126.com (M.L.); yyzdlyk@163.com (Y.Y.); todd1997@mail.nwpu.edu.cn (Z.W.); qingsong4321@126.com (Q.D.); yangxiao8792@163.com (X.Y.); 15227117791@163.com (X.G.); a_moon1096@163.com (W.L.); 2Institute of Medical Research, Northwestern Polytechnical University, Xi’an 710072, China; xjchen@nwpu.edu.cn

**Keywords:** molnupiravir, EIDD-1931, enterovirus, antiviral

## Abstract

Enterovirus infections can cause hand, foot, and mouth disease (HFDM), aseptic meningitis, encephalitis, myocarditis, and acute flaccid myelitis, leading to death of infants and young children. However, no specific antiviral drug is currently available for the treatment of this type of infection. The Unites States and United Kingdom health authorities recently approved a new antiviral drug, molnupiravir, for the treatment of COVID-19. In this study, we reported that molnupiravir (EIDD-2801) and its active form, EIDD-1931, have broad-spectrum anti-enterovirus potential. Our data showed that EIDD-1931 could significantly reduce the production of EV-A71 progeny virus and the expression of EV-A71 viral protein at non-cytotoxic concentrations. The results of the time-of-addition assay suggest that EIDD-1931 acts at the post-entry step, which is in accordance with its antiviral mechanism. The intraperitoneal administration of EIDD-1931 and EIDD-2801 protected 1-day-old ICR suckling mice from lethal EV-A71 challenge by reducing the viral load in various tissues of the infected mice. The pharmacokinetics analysis indicated that the plasma drug concentration overwhelmed the EC_50_ for enteroviruses, suggesting the clinical potential of molnupiravir against enteroviruses. Thus, molnupiravir along with its active form, EIDD-1931, may be a promising drug candidate against enterovirus infections.

## 1. Introduction

Enterovirus belongs to the *Enterovirus* genus of the *Picornaviridae* family. Enterovirus infections cause hand, foot, and mouth disease (HFMD), myocarditis, and a series of neurological complications in infants and young children worldwide [1,2]. Enteroviruses include polioviruses, echoviruses, coxsackieviruses, and numbered enteroviruses [3]. Among these species, enterovirus A71 (EV-A71), coxsackievirus A6 (CV-A6), and coxsackievirus A16 (CV-A16) have the potential to cause fatal infections, including aseptic meningitis (AM) and encephalitis [4,5]. Enterovirus D68 (EV-D68) sometimes causes severe neurological complications, such as acute flaccid myelitis (AFM) [6]. Coxsackievirus B3 (CV-B3), a cardiotropic virus, has been identified as one of the leading causes of viral myocarditis [7,8,9]. Although most enterovirus infections cause only mild and self-limiting diseases, the large number of cases and high prevalence of enterovirus infections throughout the world highlight the need for specific antiviral drugs against enteroviruses [10,11,12,13,14]. Unfortunately, there are no antiviral drugs currently approved to treat enterovirus infections. Although three inactivated monovalent EV-A71 vaccines have been widely used in the prevention of hand, foot, and mouth disease (HFDM) and some clinical trials have reported that these vaccines can provide efficient protection against EV-A71-associated HFMD, a cross-protection effect against CV-A6, CV-A10, and CV-A16 has rarely been observed [15,16,17,18]. Due to the vast number of different enteroviral serotypes, research on individual vaccines against all types of enteroviruses is not feasible. Therefore, the development of broad-spectrum antiviral drugs with activity against multiple serotypes of enteroviruses is urgently needed.

*β*-D-*N*^4^-hydroxycytidine (EIDD-1931, the active form of Molnupiravir) was designed and synthesized as a ribonucleotide analog and is incorporated into nascent viral RNAs in place of cytidine, increasing the frequency of lethal mutagenesis and thereby preventing the generation of offspring viruses [19]. EIDD-1931 was found capable of inhibiting various RNA viruses, including hepatitis C virus, influenza viruses, respiratory syncytial virus, Ebola virus, Venezuelan equine encephalitis virus, SARS-CoV-2, and human seasonal coronaviruses [19,20,21,22,23]. As a ribonucleoside analog, EIDD-1931 is quickly metabolized in the enterocytes of non-human primates after oral administration, which could reflect poor human bioavailability. To overcome this problem, EIDD-2801 (molnupiravir, prodrug of EIDD-1931) was designed. In vitro experiments demonstrated that EIDD-2801 increased oral bioavailability in non-human primates and ferrets compared to EIDD-1931 [24].

In this study, we evaluated the antiviral activity of EIDD-1931 and EIDD-2801 against enterovirus EV-A71 in vivo and in vitro, verified the stage of action of EIDD-1931 against EV-A71, and measured the broad-spectrum anti-enterovirus activities of EIDD-1931 and EIDD-2801.

## 2. Materials and Methods

### 2.1. Cells, Viruses, and Reagents

RD (Human Rhabdomyosarcoma cell) and Vero cells were purchased from the American Type Culture Collection (ATCC, CCL-136, CCL-81). Huh7 cell line was obtained from the National Infrastructure of Cell Line Resource, China. All cell lines were cultured as monolayers in Dulbecco’s modified Eagle’s medium (DMEM, Gibco, cat no. 2367374) supplemented with 10% fetal bovine serum (FBS, Gibco, cat no. 1807622) and 1% penicillin/streptomycin (PS, Gibco, cat no. 2321152) at 37 °C in the presence of 5% CO_2_. The EV-A71 H strain was purchased from the American Type Culture Collection (ATCC, VR-1432). The EV-A71 AH08/06 strain (GenBank no. HQ611148.1) was isolated from an HFMD patient during an outbreak in 2008 in Anhui, China [25]. The EV-D68 strain STL-2014–12 (GenBank no. KM881710), CV-A6 TW-2007-00141 strain (GenBank no. KR706309), and CV-A16 strain 190-D1 (GenBank no. JX127258) were provided by Professor Tong Cheng from Xiamen University [26,27,28]. All viruses were propagated and titrated in RD cells according to the previous reports [29]. The CV-B3 Nancy strain was preserved in our lab and propagated in Vero cells.

The mouse anti-EV-A71 VP1 antibody (12D7) was obtained from the laboratory of Professor Tong Cheng [30]. The anti-alpha tubulin antibody (cat no. ab7291) and the horseradish peroxidase (HRP)-conjugated goat anti-mouse IgG antibody (cat no. ab205719) were purchased from Abcam. Alexa Fluor 647 donkey anti-mouse IgG (H + L) (cat no. A31571) and Hoechst 33,342 Fluorescent Stain Solution (cat no. H21492) were obtained from Invitrogen by Thermo Fisher Scientific.

EIDD-1931(cat no. T8498) and EIDD-2801 (cat no. T8309) powders were purchased from TargetMol (Shanghai, China), and NITD008 (7-Deaza-2′-C-acetylene-adenosine, cat no. HY-12957) powders were purchased from MCE (Shanghai, China). They were dissolved in DMSO at a 100 mM stock concentration and then stored at −20 ºC for in vitro tests. For the in vivo activity evaluation, EIDD-1931 and EIDD-2801 were solubilized in saline and then subjected to alternating ultrasound treatment to ensure full dissolution.

### 2.2. Antiviral Activity Assay

The antiviral activity assay was performed in 96-well plates as previously described [31]. Briefly, RD, Vero, and Huh7 cells were seeded in a white-walled clear-bottom 96-well plate at 1.0 × 10^4^ cells/well and grown for 24 h before infection. EIDD-1931 or EIDD-2801 was added to the cells in a 3-fold dilution series (ranging from 200 to 0.01 μM) with 2% FBS DMEM, and DMSO treatment was set as a control. The virus was diluted to 100× TCID_50_ using the 2% FBS DMEM and added to the above-mentioned 96-well plates. After incubation at 37 °C for 72 h, the antiviral effects of the test compounds were measured using a CellTiter-Glo cell viability assay kit (Promega, cat no. G7570) following the manufacturer’s instructions. The luminescence was read using a SpectraMax M5 microplate reader (Molecular Devices). The half-maximal effective concentration (EC_50_) was calculated using Origin 9.0 software.

To explore the inhibitory activity of EIDD-1931 and EIDD-2801 on EV-A71, the viral loads in the supernatant and the viral RNA levels in the cell lysates were measured by TCID_50_ and quantitative real-time PCR (qRT-PCR), respectively. RD cells were seeded in 12-well plates at 4.0 × 10^5^ cells/well. After 24 h, the cells were incubated with the EV-A71 virus at a multiplicity of infection (MOI) of 0.1 PFU/mL and the diluted test compounds. After 1.5 h, the supernatant was aspirated and supplemented with 1 mL of 2% FBS DMEM containing different concentrations of test compounds. The virus particle yields in the supernatant and viral RNA in cells were quantified at 30 h post infection (h.p.i.).

### 2.3. Immunofluorescence Assay

Vero cells were seeded in 96-well plates at 1.0 × 10^4^ cells/well and grown for 24 h before infection, followed by infection with the EV-A71 H strain at an MOI of 1. The test compounds were added at the indicated concentrations. After 16 h, the cells were fixed with 4% paraformaldehyde at room temperature for 30 min and washed with PBS. After that, the cells were perforated by 0.1% Triton X-100 at room temperature for 30 min and blocked by 5% BSA at 37 °C for 30 min, followed by incubation with mouse anti-EV-A71 VP1 (12D7) antibody (1:5000) at 37 °C for 1 h and with Alexa Fluor 546 donkey anti-mouse IgG (H + L) antibody (1:500) for 1 h at 37 °C. The cells were washed three times, and the nucleus was stained using the Hoechst 33,342 fluorescent stain for 30 min at room temperature. The cell images were captured using a Leica DMi8 inverted microscope (Leica).

### 2.4. Time-of-Drug-Addition Assay

Time-of-drug-addition assay was carried out according to previous reports [32]. Briefly, RD cells were plated in 12-well plates at 4.0 × 10^5^ cells/well and incubated overnight. Then, the cells were treated with the EV-A71 H strain at an MOI of 0.01 PFU/mL and EIDD-1931 (10 μM) or NITD008 (1 μM) according to the drug addition schedule. At 30 h.p.i., the cells were harvested, and total RNA was extracted then subjected to qRT-PCR.

### 2.5. Quantitative Real-Time PCR

Total cellular RNA was extracted from the cell samples using an RNeasy mini kit (Qiagen, Hilden, Germany, cat no. 74106). Quantitative real-time PCR (qRT-PCR) was performed using the One Step PrimeScript RT-PCR Kit (TaKaRa, Otsu, Japan, cat no. RR064A) according to the manufacturer’s instructions. The details of the primer and probe sequences specific to the EV-A71 virus are as follows: forward primer, 5′-CCAATCTCAGCGGCTTGGAG-3′; reverse primer, 5′-CACTCAAGCTCTACCGGCAC-3′; and probe, FAM-TCCAATCGATGGCTGCTCACCTGCGT-BHQ1. Virus RNA copy numbers were calculated by comparing the cycle threshold (Ct) value obtained from each sample with that of the standard curve based on known copy numbers.

### 2.6. Western Blot Assay

RD cells were seeded in 12-well plates at 4.0 × 10^5^ cells/well, incubated overnight, and then supplemented with 1 mL of 2% FBS DMEM containing EV-A71 virus at an MOI of 0.1 and the test compound (EIDD-1931: ranging from 10 to 0.63 μM; EIDD-2801: ranging from 100 to 6.3 μM). The cell control group was supplemented with 1 mL of 2% FBS DMEM and incubated for 1.5 h at 37 °C; the cells were then washed three times with PBS and supplemented with 1 mL of 2% FBS DMEM. After 24 h, the cells were washed three times with cold PBS, followed by addition of 150 μL of RIPA lysis buffer (APPLYGEN, cat no. 1053) containing protease and phosphatase inhibitor cocktail (Thermo, cat no. A32961). The proteins were separated on a 10% SDS-PAGE gel and then transferred onto a PVDF membrane. The PVDF membranes were blocked using 5% skim milk for 1 h at room temperature and then incubated with anti-EV-A71 VP1 antibody 12D7 (1:1000) or anti-alpha tubulin antibody (1:5000) as a control at 4 °C overnight. The membranes were washed three times and incubated with horseradish peroxidase (HRP)-conjugated goat anti-mouse IgG antibody (1:5000) at room temperature for 1 h. Finally, the protein bands were imaged using a fluorchem imaging system (ProteinSimple).

### 2.7. EV-A71 Infection in Mice

The in vivo therapeutic efficacy of EIDD-1931 and EIDD-2801 was evaluated using 1-day-old suckling ICR mice as previously described [30]. Briefly, groups of 1-day-old suckling mice (purchased from the Beijing Vital River Laboratory, China) were intraperitoneally (i.p.) inoculated with 25 μL of the EV-A71 H strain (10^6^ PFU per mouse). After 4 h post infection, EIDD-1931 (200, 66, 22, and 7.41 mg/kg) and EIDD-2801 (200, 66, 22, and 7.41 mg/kg) were i.p. administered. Saline was administered for the virus control group. The treatment was continued once a day for 7 consecutive days. The mice were monitored daily for weight and mortality until 21 d.p.i. The survival rate was processed using the log-rank test.

Another three groups of 1-day-old ICR suckling mice (11 newborn mice per group) were infected with 25 μL of EV-A71 H strain virus (10^6^ PFU per mouse) intraperitoneally. After 4 h post infection, EIDD-1931 (200 mg/kg), EIDD-2801 (200 mg/kg), and saline were i.p. administered. The treatment was continued daily for 4 days. The mice were then euthanized, and the brain, heart, liver, intestines, lung, and limb muscles were separately harvested. The levels of viral RNA in each tissue were measured by qRT-PCR.

### 2.8. Ethics Statement

All animal experiments were approved by the Institutional Animal Care and Use Committee of the Beijing Institute of Pharmacology and Toxicology. All work with infectious viruses was performed at the biosafety level 2 (BSL-2) or animal biosafety level 2 laboratory (ABSL-2).

### 2.9. Statistical Analysis

Statistical analyses were carried out using the GraphPad Prism 7.0 software. A log-rank test was performed for the survival analysis. An unpaired two-tailed Student’s *t* test, or a one-way analysis of variance was used to analyze the statistical significance of two or multiple groups, respectively. For each test, *p* < 0.05 was considered to indicate a statistically significant difference.

## 3. Results

### 3.1. EIDD-1931 and EIDD-2801 Inhibit EV-A71 Infection In Vitro

To determine the inhibitory activity of EIDD-1931 and EIDD-2801 against EV-A71 virus, a cytopathic effect (CPE) protection assay was carried out using different cell lines. As shown in Figure 1B–G, EIDD-1931 and EIDD-2801 both exhibited a steady CPE protection potential in multiple cell lines infected with EV-A71 virus in a dose-dependent manner. The half-maximal effective concentrations (EC_50_) value of EIDD-1931 against EV-A71 virus was 5.13 ± 0.56 µM in RD cells, 7.04 ± 0.38 µM in Vero cells, and 4.43 ± 0.33 µM in Huh-7 cells. The EC_50_ value of EIDD-2801 against EV-A71 virus was 70.12 ± 4.40 µM in RD cells, 88.52 ± 3.18 µM in Vero cells, and 35.64 ± 0.47 µM in Huh7 cells. The half-cytotoxic concentrations (CC_50_) value of EIDD-1931 was 80.47 ± 0.02 µM in RD cells, 14.07 ± 0.43 µM in Vero cells, and 34.09 ± 0.06 µM in Huh7 cells. However, no significant cytotoxicity was observed for EIDD-2801 in all tested cell lines under 100 µM. The select index (SI) of EIDD-1931 was 15.69 in RD cells, 2.0 in Vero cells, and 7.69 in Huh7 cells. The select index (SI) of EIDD-2801 was >1.43 in RD cells, >1.13 Vero cells, and >2.81 in Huh7 cells. The above CPE protection results suggested that the in vitro antiviral activity of EIDD-1931 was higher than that of EIDD-2801, which was consistent with our expectations.

To further explore the inhibitory efficacy of EIDD-2801 and EIDD-1931 on viral RNA replication and infectious viral particle generation, a qRT-PCR assay and a PFU assay were conducted, respectively. Treatment with EIDD-1931 or EIDD-2801 inhibited the replication of viral RNA in a dose-dependent manner (Figure 2B,D), and the propagation of infectious viral particles was also inhibited (Figure 2A,C). EIDD-1931 seems to be 10 times more potent than EIDD-2801, which is in accordance with the EC_50_ results.

### 3.2. EIDD-1931 and EIDD-2801 Reduce the Production of EV-A71 Virus Protein

To explore the potential of EIDD-1931 and EIDD-2801 for inhibiting EV-A71 virus protein production, we detected viral capsid protein (VP1) levels using immunofluorescence staining and Western blot assays. Fluorescence imaging showed that EIDD-1931 and EIDD-2801 strongly inhibited the production of VP1 proteins. As shown in Figure 2E, the expression of EV-A71 virus VP1 protein was almost completely blocked by EIDD-1931 at a concentration of 10 µM and partly inhibited by EIDD-2801 at a concentration of 100 µM, which is in accordance with the observations in the CPE inhibition and viral yield reduction assays. The results of Western blot revealed that the expression levels of viral VP1 proteins were significantly attenuated in the presence of EIDD-1931 or EIDD-2801 in a dose-dependent manner (Figure 2F–I). Taken together, these data indicated that EIDD-1931 and EIDD-2801 can significantly reduce the expression of viral VP1 protein.

### 3.3. EIDD-1931 Acts at the Post-Entry Stage of EV-A71 infection

As a nucleoside analog, EIDD-1931 has been reported to be incorporated into nascent viral RNAs in place of cytidine, followed by an increasing frequency of lethal mutagenesis and a reduction in the generation of offspring virus. To determine which stage of the EV-A71 life cycle was affected by EIDD-1931, we performed a time-of-addition assay via incubation with the test compound at different time intervals according to the illustration in Figure 3A. EIDD-1931 and NITD008 belong to nucleoside analogue inhibitors, which have similar antiviral mechanism. NITD008 is not only a classical inhibitor against flavivirus infection, but also has potent anti-EV-A71 activity as reported in multiple literatures [33,34,35]. Therefore, we used it as the positive control drug for viral suppression. As expected, EIDD-1931 inhibited EV-A71 viral RNA production at stages III and IV, which indicates that EIDD-1931 acts at the post-entry stage of virus infection. No inhibition effects were observed at the pre-incubation, attachment, or internalization phases of the viral life cycle (Figure 3B). To sum up, these data suggested that EIDD-1931 acts at the post-entry step.

### 3.4. EIDD-1931 and EIDD-2801 Protected 1-Day-Old ICR Suckling Mice from Lethal EV-A71 Challenge

To test the in vivo therapeutic efficacy of EIDD-1931 and EIDD-2801 against EV-A71 infection, we used a lethal 1-day-old ICR suckling mouse model for evaluation of antiviral activity [36]. After the mice were intraperitoneally (i.p.) infected with 10^6^ PFU of EV-A71 virus (H strain), different concentrations of EIDD-1931 (200, 66.6, 22.2, and 7.41 mg/kg) and EIDD-2801 (200, 66.6, 22.2, and 7.41 mg/kg) were administered i.p. for 7 consecutive days. Body weight and survival were observed until 21 d.p.i. The mice receiving a dose of 200 and 66.6 mg/kg of EIDD-1931 all survived and generally exhibited good weight maintenance and growth. Additionally, 60% survival protection was observed in the mice that received EIDD-1931 (22.2 mg/kg), whereas those within the vehicle group or the low-dose group (7.41 mg/kg) died within 12 days (Figure 4A,B). Meanwhile, 100% and 67% survival rates were observed in the 200 and 66.67 mg/kg of EIDD-2801 treatment groups, respectively, compared with the vehicle group (Figure 4C). Collectively, EIDD-1931 and EIDD-2801 both exhibited potent in vivo antiviral activity against EV-A71 infection.

### 3.5. EIDD-1931 and EIDD-2801 Reduce the Viral Loads in Various Tissues of EV-A71-Infected Mice

To further characterize the in vivo antiviral effects of EIDD-1931 and EIDD-2801, we determined the viral loads of various tissues in EV-A71-infected mice treated with the test compounds or vehicle. After the mice were infected with 106 PFU of EV-A71 virus (H strain), EIDD-1931 and EIDD-2801 at a dosage of 200 mg/kg or vehicle were administered via i.p. for 4 consecutive days. Then, the mice were dissected and several tissues, including brain, heart, intestine, liver, limb muscle, and lung, were collected to determine the viral RNA load using qRT-PCR. As shown in Figure 5, treatment with EIDD-1931 and EIDD-2801 significantly reduced the viral loads in the above tissues. These results suggested that EIDD-1931 and EIDD-2801 potently inhibit the viral replication of EV-A71 in mouse models.

### 3.6. EIDD-1931 and EIDD-2801 Have Broad-Spectrum Antiviral Activity against Multiple Enteroviruses In Vitro

To identify whether EIDD-1931 and EIDD-2801 have broad-spectrum anti-enterovirus activities, a CPE protection assay was performed using different species of enteroviruses, including EV-A71 AH08/06 strain, EV-D68 STL-2014–12 strain, CV-A6 TW-2007-00141 strain, CV-A16 190-D1 strain, and CV-B3 Nancy strain. The results demonstrated that EIDD-1931 significantly inhibits infections by multiple enteroviruses, with EC_50_ values of <20 μM, whereas EIDD-2801 showed relatively low inhibitory activity against EV-V68, CV-A16, and CV-B3, and no inhibitory activity against CV-A6. Additionally, we determined the antiviral activity of EIDD-1931 and EIDD-2801 against EV-A71 clinical isolate strain AH 08/06, representing EC_50_ values of 2.79 ± 0.89 and 30.29 ± 0.55 µM, respectively, with corresponding SI values of 12.22 and >3.3 (Table 1). These results indicated that EIDD-1931 and EIDD-2801 exhibit broad-spectrum in vitro anti-enterovirus activity.

## 4. Discussion

Enteroviruses are diverse and spread easily. The epidemics and outbreaks of diseases caused by enteroviruses represent one of the most important public health problems all over the world that seriously threatens the health of children. Hence, effective antiviral treatment against these viruses is urgently needed.

As a key virus-encoded enzyme in the viral replication cycle, RNA-dependent RNA polymerase (RdRp) plays an important role in viral genome replication. Considering the similar structure of RdRps in different viruses and the conservation of their active elements, RdRps have become one of the best targets for developing broad-spectrum antiviral agents. EIDD-1931, as a ribonucleotide analog, has been established as an active agent against various RNA viruses in vitro. An in vivo activity study demonstrated that EIDD-1931 can be efficacious against respiratory syncytial virus and both seasonal and highly pathogenic avian influenza A virus in mouse models, reducing lung virus loads and alleviating disease biomarkers via oral (p.o.) administration [20]. Although EIDD-1931 exhibits good oral bioavailability in rodents, some researchers found that the plasma concentrations were low in cynomolgus macaques, which reflects poor bioavailability in humans. To overcome this problem, molnupiravir (EIDD-2801, the pro-drug of EIDD-1931) was designed. The data for EIDD-2801 showed that it has increased oral bioavailability in non-human primates and ferrets compared to EIDD-1931, but both have similar oral bioavailability in mice [24].

In this study, we reported that EIDD-1931 and EIDD-2801 have broad-spectrum anti-enterovirus potential in vitro. Further research on the mechanism of EIDD-1931 found that it acts at the post-entry step. However, the post-entry step includes virus replication, virus assembly, and virus release. EIDD-1931, as a ribonucleotide analog, can introduce lethal mutagenesis during viral RNA replication [19]. We speculate that EIDD-1931 acts at the replication stage of EV-A71. To verify this possibility, additional replicon experiments need to be performed. Because the enterovirus mainly infects infants and young children, we developed and tested a corresponding 1-day-old suckling mouse model. The in vivo data suggested that EIDD-1931 and EIDD-2801 have therapeutic significance in lethal EV-A71-infected mice; when administered therapeutically for 4 consecutive days, they significantly reduced the viral load in various tissues of EV-A7-infected mice. The pharmacokinetic profile is an important reference for estimating clinical efficacy. The Cmax of EIDD-2801 was 16.13 μM following oral administration of a dose of 800 mg twice daily for 5 days, which is greater than the EC_50_ of enteroviruses (the in vivo metabolite of EIDD-2801 was EIDD-1931), indicating that EIDD-2801 is a potential countermeasure against enteroviruses [37].

In conclusion, EIDD-1931 and EIDD-2801 exhibited powerful anti-enterovirus effects in vitro and in vivo and demonstrated excellent broad-spectrum anti-enterovirus activity in this study. Combined with complete in vivo data and the high safety of EIDD-2801, we believe that EIDD-2801 and its active form, EIDD-1931, may be a promising drug candidate against enterovirus infections. Although molnupiravir has obtained emergency authorization to treat SARS-CoV-2 infection in many countries, it is currently only approved for young adults over 18 years old. The enterovirus mainly infects children under 5 years old. Therefore, the potential risk of clinical application of molnupiravir in the treatment of enterovirus infection needs to be further studied.

## Figures and Tables

**Figure 1 viruses-14-01142-f001:**
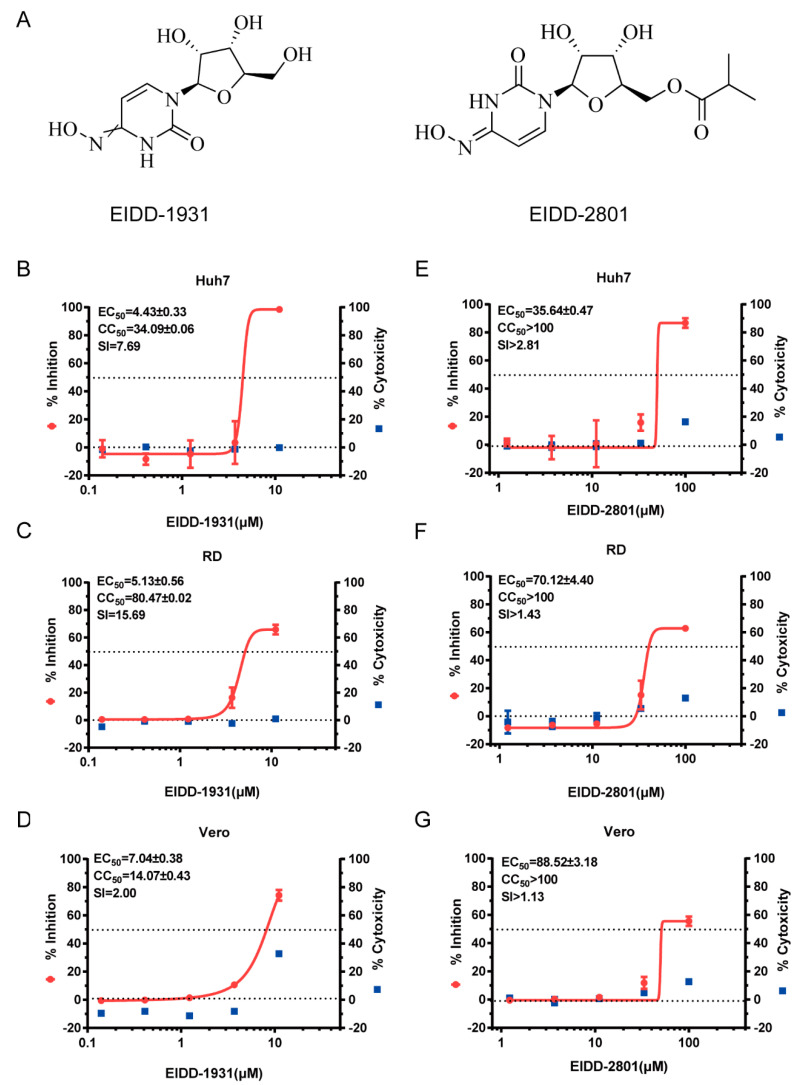
The molecular formula of EIDD−1931 and EIDD−2801 and the in vitro antiviral effects against EV−A71. (**A**) The molecular formula of EIDD−1931 and EIDD−2801. (**B**–**G**) The antiviral activities of EIDD−1931 (**B**–**D**) and EIDD2801 (**E**–**G**) against EV−A71 in different cells lines. RD cells, Vero cells, and Huh7 cells were infected with the EV−A71 H strain at 100 × TCID_50_. Different doses of the test compounds were then added. At 72 h.p.i, the antiviral parameters were measured. The antiviral effects and cytotoxicity of EIDD−1931 and EIDD−2801 were measured using a CellTiter−Glo cell viability assay kit. The EC_50_ and CC_50_ were calculated using Origin 9.0 software. SI = CC_50_/IC_50_.

**Figure 2 viruses-14-01142-f002:**
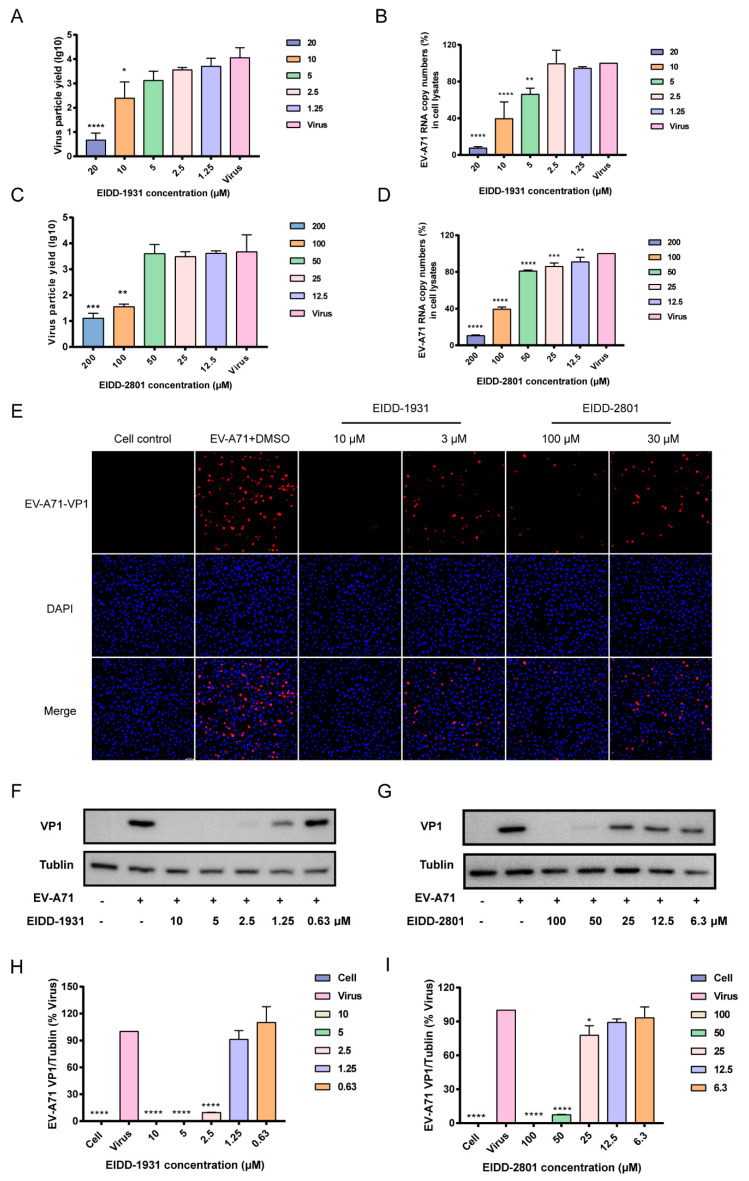
Antiviral activities of EIDD−1931 and EIDD−2801 against EV−A71 in vitro. (**A**–**D**) EV−A71 virus particle yields and viral RNA reduction assay. RD cells were infected with EV71 at an MOI of 0.1 PFU/mL in the presence of different concentrations of EIDD−1931 or EIDD−2801. After 30 h, the virus particle yields in the supernatant were measured by TCID_50_. Total cellular RNA was extracted and subjected to qRT−PCR to measure viral RNA expression. All data are shown as means ± standard deviation from three independent experiments. Statistical significance was calculated using one-way ANOVA. * *p* < 0.05, ** *p* < 0.01, *** *p* < 0.001, and **** *p* < 0.0001. (**E**–**G**) Inhibitory effects of EIDD-1931 and EIDD−2801 on viral VP1 protein. (**E**) RD cells were infected with EV−A71 at an MOI of 1 PFU/mL in the presence of EIDD−1931 or EIDD−2801. At 16 h.p.i., the cells were fixed for immunofluorescence assays. EV−A71 VP1 proteins were stained using mouse anti−EV−A71 antibody (12D7) (red), and cell nuclei were stained using Hoechst 33,342 (blue), scale bar: 50 µm. (**F**,**G**) RD cells infected with EV−A71 (MOI = 1) were treated with the test compounds at the indicated concentrations. At 24 h post infection, the cells were collected for Western blot analysis using anti−EV−A71VP1 antibodies (12D7). (**H**,**I**) Proteins levels of EV−A71 VP1 were quantified by ImageJ software. * *p* < 0.05, **** *p* < 0.0001.

**Figure 3 viruses-14-01142-f003:**
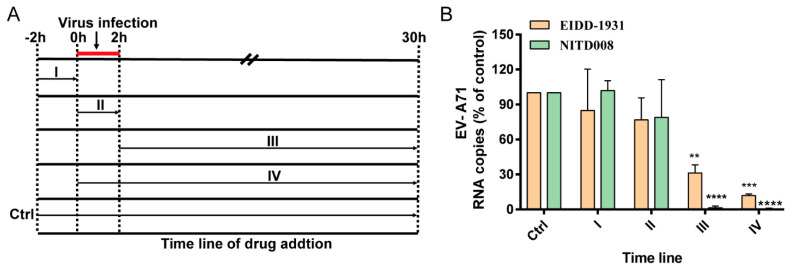
EIDD−1931 acts at the post-entry stage of EV−A71 virus infection. (**A**) Schematic of the time-of-drug-addition assay. (**B**) Levels of intracellular viral RNA. RD cells were treated with EV−A71 and the test compounds according to the timeline shown in panel A. After 30 h, the viral RNA was extracted and determined via qRT−PCR. The data collected from three independent experiments are represented as means ± standard deviations. Statistical significance was analyzed using one-way ANOVA. ** *p* < 0.01, *** *p* < 0.001, and **** *p* < 0.0001.

**Figure 4 viruses-14-01142-f004:**
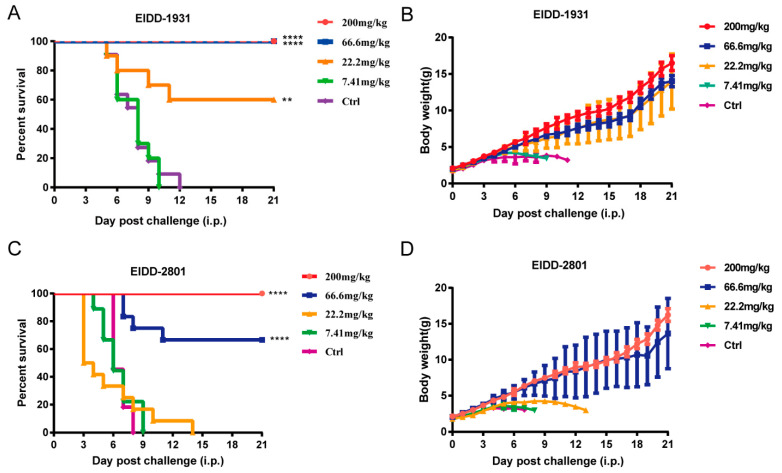
EIDD-1931 and EIDD-2801 exert in vivo antiviral effects against EV-A71 infection. Four groups of 1-day-old ICR suckling mice were i.p. infected with 10^6^ PFU of EV-A71 virus (H strain), followed by i.p. treatment with EIDD-1931, EIDD-2801, or vehicle at the indicated dosages for 7 consecutive days. Survival (**A**,**C**) and body weight (**B**,**D**) were recorded every day until 21 d.p.i. Data for B and D are presented as means ± standard deviations. Survival data were analyzed with a log-rank test. ** *p* < 0.01, **** *p* < 0.0001.

**Figure 5 viruses-14-01142-f005:**
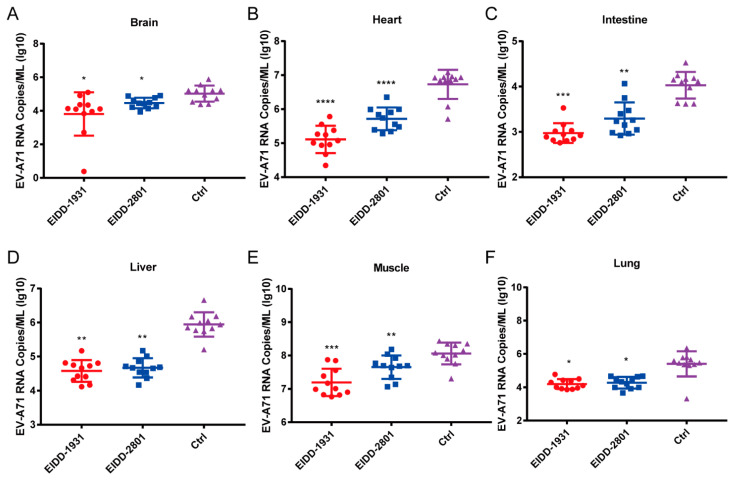
EIDD-1931 and EIDD-2801 reduce the viral loads in various tissues of 1-day-old ICR suckling mice. Three groups of 1-day-old ICR suckling mice were i.p. infected with 10^6^ PFU of EV-A71 virus (H strain), followed by i.p. treatment with EIDD-1931 and EIDD-2801 at a dosage of 200 mg/kg for 4 consecutive days. Then, the mice were euthanized, and brain (**A**), heart (**B**), intestine (**C**), liver (**D**), limb muscles (**E**) and lung (**F**) were separately harvested to determine the viral RNA loads using qRT-PCR. Data are presented as means ± standard deviations, and Student’s unpaired *t* test was performed for statistical analysis. * *p* < 0.05, ** *p* < 0.01, *** *p* < 0.001, and **** *p* < 0.0001.

**Table 1 viruses-14-01142-t001:** Broad-spectrum anti-enterovirus potency of EIDD-1931 and EIDD-2801.

Enterovirus	Cell Line	EIDD-1931		EIDD-2801	
		EC_50_ (μM)	SI	EC_50_ (μM)	SI
EV-A71 AH08/06	Huh7	2.79 ± 0.89	12.22	30.29 ± 0.55	>3.3
EV-D68 STL-2014–12	RD	9.69 ± 0.04	8.3	86.21 ± 9.17	>1.6
CV-A6 TW-2007-00141	RD	15.72 ± 0.38	5.11	>100	-
CV-A16 190-D1	RD	4.14 ± 0.27	19.43	44.91 ± 4.23	>2.23
CV-B3 Nancy	RD	3.65 ± 0.82	3.85	87.23 ± 10.84	>1.15

The EC_50_ values were determined in three independent experiments, and the data are presented as the means ± standard deviations. The select index (SI) is the ratio of CC_50_ to EC_50_.

## Data Availability

All data are available in the main text.

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
