# Peer review of "Molnupiravir and Its Active Form, EIDD-1931, Show Potent Antiviral Activity against Enterovirus Infections In Vitro and In Vivo"

_viruses, 2022, doi:10.3390/v14061142_

Round 1

Reviewer 1 Report

Authors studied the β-D-N4-hydroxycytidine (EIDD-1931, the active form of Molnupiravir) that is  promising drug candidate for the therapy of enterovirus infections.

Major remark:

Why does authors choose NITD008 (7-Deaza-2'-C-acetylene-adenosine, cat 93 no. HY-12957) as drug control? By information from the site (https://www.medchemexpress.com/NITD008.html) NITD008 is a potent and selective flavivirus inhibitor. But in the current research were studied drug against enteroviruses (Picornaviridae family). Wouldn't it be more appropriate to use for example Vapendavir, that may inhibit EV-A71 and EV-D68 (Tijsma A et al. 2014, Anasir M. et al., 2021)?

Minor remarks:

Line 72 -- Decryption of RD cell

Line 114 -- Decryption of MOI

Line 307 --  Figure 5 “…1-year-old ICR  suckling mice…” >> should be 1-day-old

Line 308 – the same

Reviewer 2 Report

In this study, Li et al. examined the therapeutic potential of EIDD-1931and its prodrug EIDD-2801 (molnupiravir) against enterovirus infection in vitro and in vivo mouse model. They found EIDD-1931 has a higher potency than EIDD-2801 in vitro and in vivo. Their time of addition experiment suggested the potential action point of these drugs at the early stage of viral replication, most likely at the stage of RNA replication as expected from the chemical nature of the drugs. They examined several enteroviruses including clinical isolates and found these drugs showed a broad spectrum of antiviral effect. As molnupiravir has been recently approved for human use in the treatment of SARSCoV-2 infection, their finding suggested that the potential application of these drugs in treatment of enteroviral infection in human. Authors should address the following issues.

Comments

Although the study was well designed and the data supported their argument, the safety index is relatively narrow and there is potential risk of these drugs in application in young children (molnupiravir is indeed currently approved only for the treatment of COVID-19 in young adults over 18 years old). Therefore, authors should mention this potential drawback of the drugs in real world application.

Minor points

  1. In line 271, “Protecte” should be “Protected”.
  2. In Fig.4A, where is the survival curve for the dose of 66.6mg/kg? Is it completely overlapped with the line of other doses?
  3. What was the cell line used in the experiments shown in Table 1? It should be described in the legend for clarity.

Reviewer 3 Report

In this manuscript authored by Li et al., the antiviral activity of molnupiravir (EIDD-2801) and its active form, EIDD-1931 was evaluated in cell and in vivo against a group of enteroviruses. Furthermore, the authors provided evidence suggest that these two compound functions at the step post viral entry. Overall, this study is reasonable and interesting.

Minor comments:

  1. Did EIDD-2801 or EIDD-2801 treatment change the morphology of viral particle of EV71?
  2. It would be better to perform quantitation analysis on Figure 2E-2G.
  3. Why chose the concentrations 200, 66.6, 22.2, and 7.41 mg/kg in vivo? Please clarify.
  4. One-way ANOVA may be more suitable to analyze the data of Figure 5.
  5. The SIs of EIDD-1931 for most enteroviruses are below 10, suggesting that its board-spectrum antiviral activity is NOT potent. The author should discuss this limitation.

Round 2

Reviewer 3 Report

The authors have addressed my concerns.